# DIFFERENTIABLE SPATIAL PLANNING USING TRANSFORMERS

## ABSTRACT

We consider the problem of spatial path planning. In contrast to the classical solutions which optimize a new plan from scratch and assume access to the full map with ground truth obstacle locations, we learn a planner from the data in a differentiable manner that allows us to leverage statistical regularities from past data. We propose Spatial Planning Transformers (SPT), which given an obstacle map learns to generate actions by planning over long-range spatial dependencies, unlike prior data-driven planners that propagate information locally via convolutional structure in an iterative manner. In the setting where the ground truth map is not known to the agent, we leverage pre-trained SPTs to in an end-to-end framework that has the structure of mapper and planner built into it which allows seamless generalization to out-of-distribution maps and goals. SPTs outperform prior state-of-the-art across all the setups for both manipulation and navigation tasks, leading to an absolute improvement of 7-19%.

## 1 INTRODUCTION

The problem of path planning has been a bedrock of robotics. Given an obstacle map of an environment and a goal location in the map, the task is to output a shortest path to the goal location starting from any position in the map. We consider path planning with spatial maps. Building a top-down spatial map is common practice in robotic navigation as it provides a natural representation of physical space (Durrant-Whyte & Bailey, 2006). In fact, even robotic manipulation can also be naturally phrased via spatial map using the formalism of configuration spaces (Lozano-Perez, 1990), as shown in Figure 1. This problem has been studied in robotics for several decades, and classic goto planning algorithms involve Dijkstra et al. (1959), PRM (Kavraki et al., 1996), RRT (LaValle & Kuffner Jr, 2001), RRT* (Karaman & Frazzoli, 2011), etc.

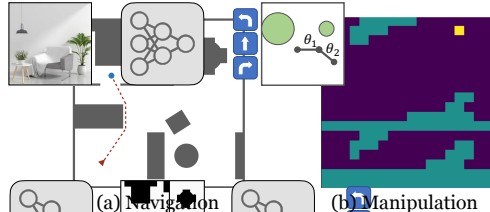

(a) Navigation    (b) Manipulation

**Figure 1:** Spatial Path Planning: The raw observations (top left) and obstacles can be represented spatially via top-down map in navigation (left) and via configuration space in manipulation (right).

Our objective is to develop methods that can *learn* to plan from data. However, a natural question is why do we need learning for a problem which has stable classical solutions? There are two key reasons. First, classical methods do not capture statistical regularities present in the natural world, (for e.g., walls are mostly parallel or perpendicular to each other), because they optimize a plan from scratch for each new setup. This also makes analytical planning methods to be often slow at inference time which is an issue in dynamic scenarios where a more reactive policy might be required for fast adaptation from failures. A learned planner represented via a neural network can not only capture regularities but is also efficient at inference as the plan is just a result of forward-pass through the network. Second, a critical assumption of classical algorithms is that a global ground-truth obstacle space must be known to the agent ahead of time. This is in stark contrast to biological agents where cognitive maps are not pixel-accurate ground truth location of agents, but built through actions in the environment, e.g., rats build an implicit map of the environment incrementally through trajectories enabling them to take shortcuts (Tolman, 1948). A learned solution could not only provides the ability to deal with partial, noisy maps and but also help build maps on the fly while acting in the environment by backpropagating through the generated long-range plans.

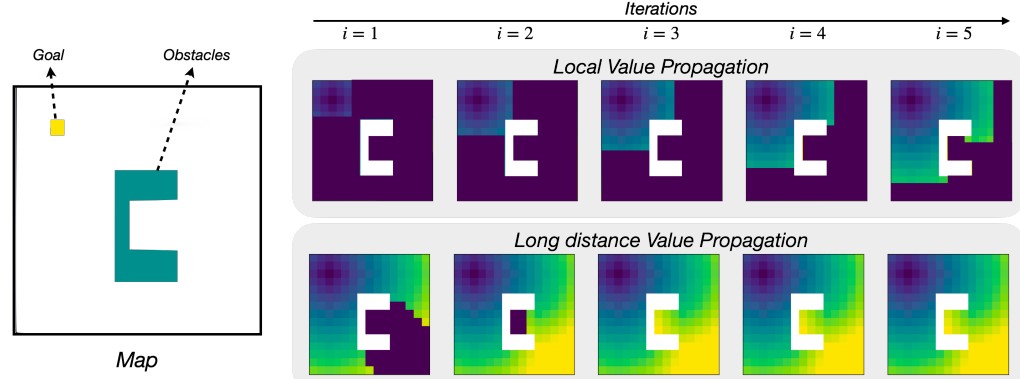

**Figure 2: Local vs Long-distance value propagation.** Figure showing an example of number of iterations required to propagate distance values over a map using local and long-distance value propagation. The obstacle map and goal location shown on the left and the distance value predictions over 5 iterations is shown on the right (distance values increase from blue to yellow). Prior methods based on convolutional networks use local value propagation and require many iterations to propagate values accurately over the whole map (top right). Our method is based on long-distance value propagation between points without any obstacle between them. This type of value propagation can cover the whole map in 3 iterations in this example (bottom right).

Several recent works have proposed data-driven path planning models (Tamar et al., 2016; Karkus et al., 2017; Nardelli et al., 2018; Lee et al., 2018). Similar to how classical algorithms, like Dijkstra et al. (1959), move outward from the goal one cell at a time to predict distances iteratively based on the obstacles in the map, current learning-based spatial planning models propagate distance values in only a *local* neighborhood using convolutional networks. This kind of local value propagation requires $\mathcal{O}(M)$ iterations, where $M$ is the map dimension. In theory, however, the optimal paths can be computed much more efficiently with total iterations that are on the order of number of obstacles rather than the map size. For instance, consider two points with no obstacle between, an efficient planner could directly connect them with interpolated distance. Nonetheless, this is possible only if the model can perform long-range reasoning in the obstacle space which is a challenge.

In this work, our goal is to capture this *long-range* spatial relationship. Transformers (Vaswani et al., 2017) are well suited for this kind of computation as they treat the inputs as sets and propagate information across all the points within the set. Building on this, we propose *Spatial Planning Transformers* (SPT) which consists of attention heads that can attend to any part of the input. The key idea behind the design of the proposed model is that value can be propagated between distant points if there are no obstacles between them. This would reduce the number of required iterations to $\mathcal{O}(n_O)$ where $n_O$ is the number of obstacles in the map. Figure 2 shows a simple example where long-distance value propagation can cover the entire map within 3 iterations while local value propagation takes more than 5 iterations – this difference grows with the complexity of the obstacle space and map size. We compare the performance of SPTs with prior state-of-the-art learned planning approaches, VIN (Tamar et al., 2016) and GPPN (Lee et al., 2018), across both navigation as well as manipulation setups. SPTs achieve significantly higher accuracy than these prior methods for the same inference time and show over $10\%$ absolute improvement when the maps are large.

Next, we turn to the case when the map is not known apriori. This is a practical setting when the agent either has access to a partially known map or just know it through the trajectories. In psychology, this is known as going from *route* knowledge to *survey* knowledege (Golledge et al., 1995) where animals aggregate the knowledge from trajectories into a cognitive map. We operationalize this setup by formulating an end-to-end differentiable framework, which in contrast to having a generic parametric policy learning (Glasmachers, 2017), has the structure of mapper and planner built into it. We first pre-train the SPT planner to capture a generic data-driven prior, and then backpropagate through it to learn a mapper that maps raw observations to obstacle map. This allows us to learn without requiring map supervision or interaction. Learned mapper and planner not only allow us to plan for new goal locations at inference but also generalize to unseen maps.

Our end-to-end mapping and planning approach provides a unified solution for both navigation and manipulation. We perform thorough experiments in both manipulation and real-world navigation maps as well as manipulation. Our approach outperforms prior state-of-the-art by a margin on both mapping and planning accuracy without assuming access to the map at training or inference.

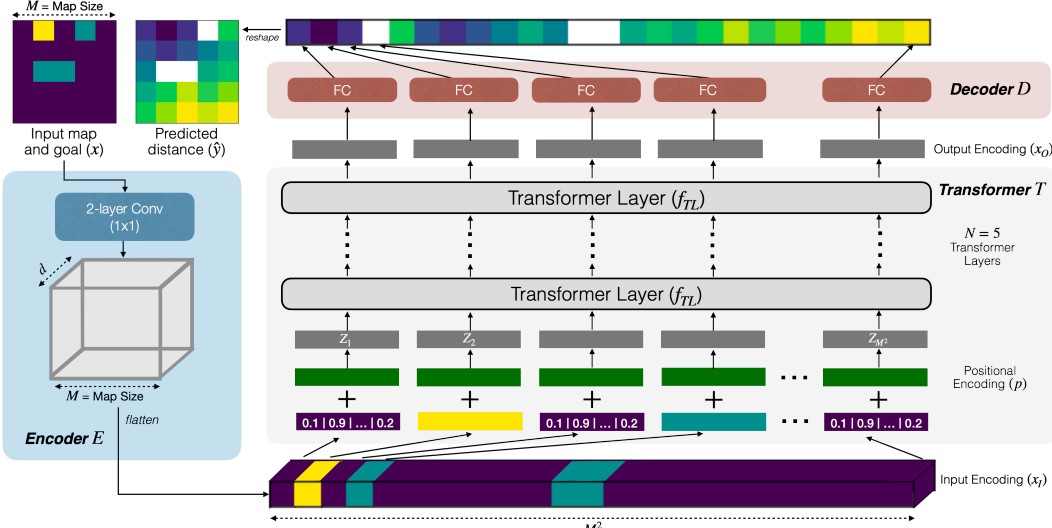

**Figure 3: Spatial Planning Transformer (SPT).** Figure showing an overview of the proposed Spatial Planning Transformer model. It consists of 3 modules: an *Encoder E* to encode the input, a *Transformer* network $T$ responsible for planning, and a *Decoder D* decoding the output of the Transformer into action distances.

## 2 PRELIMINARIES AND PROBLEM DEFINITION

We represent the input spatial map as a matrix, $m$, of size $M \times M$ with each element being 1, denoting obstacles, or 0, denoting free space. The goal location is also represented as a matrix, $g$, of size $M \times M$ with exactly one element being 1, denoting the goal location, and rest 0s. The input to the spatial planning model, $x$, consists of matrices $m$ and $g$ stacked, $x = [m, g]$, where $x$ is of size $2 \times M \times M$. The objective of the planning model is to predict $y$ which is of size $M \times M$, consisting of action distances of corresponding locations from the goal. Here, action distance is defined to be the minimum number of actions required to reach the goal.

For navigation, $m$ is a top-down obstacle map, and $g$ represents the goal position on this map. For manipulation, $m$ represents the obstacles in the configuration space of 2-dof planar arm with joint angles denoted by $\theta_1$ and $\theta_2$. Each element $(i, j)$ in $m$ indicate whether the configuration of the arm with joint angles $\theta_1 = i$ and $\theta_2 = j$, would lead to a collision. $g$ represents the goal configuration of the arm. In the first set of experiments, we will assume that $m$ is known and in the second set of experiments, $m$ is not known and the agent receives observations, $o$, from its sensors instead.

## 3 METHODS

We design a spatial planning model capable of long-distance information propagation. We first describe the design of this spatial planning module, called Spatial Planning Transformer(SPT), shown in Figure 3, which takes in a map and a goal as input and predict the distance to the goal from all locations. We then describe how the SPT model can be used as a planning module to train end-to-end learning models, which take in raw sensory observations and goal location as input and predict action distances without having access to the map.

### 3.1 SPT: SPATIAL PLANNING TRANSFORMERS

To propogate information over distant points, we use the Transformer (Vaswani et al., 2017) architecture. The self-attention mechanism in a Transformer can learn to attend to any element of the input. The allows the model to learn spatial reasoning over the whole map concurrently. Figure 3 shows an overview of the SPT model, which consists of three modules, an *Encoder E* to encode the input, a *Transformer* network $T$ responsible for spatial planning, and a *Decoder D* decoding the output of the Transformer into action distances.

**Encoder.** The Encoder $E$ computes the encoding of the input $x$: $x_I = E(x)$. The input $x \in \{0, 1\}^{2 \times M \times M}$ consisting of the map and goal is first passed through a 2-layer convolutional network (LeCun et al., 1998) with ReLU activations to compute an embedding for each input element. Both layers have a kernel size of $1 \times 1$, which ensures that the embedding of all the obstacles is

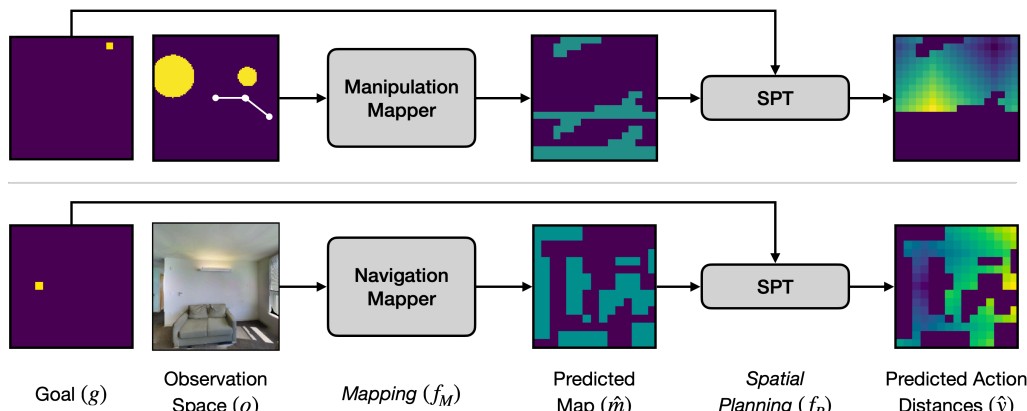

| Goal ($g$) | Observation Space ($o$) | *Mapping* ($f_M$) | Predicted Map ($\hat{m}$) | *Spatial Planning* ($f_P$) | Predicted Action Distances ($\hat{y}$) |

**Figure 4: End-to-end Mapping and Planning.** Figure showing an overview of end-to-end mapping and planning model for both the navigation and manipulation tasks.

identical to each other, and the same holds true for free space and the goal location. The output of this convolutional network is of size $d \times M \times M$, where $d$ is the embedding size. This output is then flattened to get $x_I$ of size $d \times M^2$ and passed into the Transformer network.

**Transformer.** The Transformer network $T$ converts the input encoding into the output encoding: $x_O = T(x_I)$. It first adds the positional encoding to the input encoding. The positional encoding enables the Transformer model to distinguish between the obstacles at different locations. We use a constant sinusoidal positional encoding (Vaswani et al., 2017):

$$p_{(2i,j)} = \sin(j/C^{2i/d}), \qquad p_{(2i+1,j)} = \cos(j/C^{2i/d})$$

where $p \in \mathcal{R}^{d \times M^2}$ is the positional encoding, $j \in \{1, 2, \ldots, M^2\}$ is the position of the input, $i \in \{1, 2, \ldots, d/2\}$, and $C = M^2$ is a constant.

The positional encoding of each element is added to their corresponding input encoding to get $Z = x_I + p$. $Z$ is then passed through $N = 5$ identical Transformer layers ($f_{\text{TL}}$) to get $x_O$.

**Decoder.** The Decoder $D$ computes the distance prediction $\hat{y}$ from $x_O$ using a position-wise fully connected layer:

$$\hat{y}_i = W_D^T x_{T,i} + b_D$$

where $x_{T,i} \in \mathcal{R}^{d \times 1}$ is the input at position $i \in 1, 2, \ldots, M^2$, $W_D \in \mathcal{R}^{d \times 1}, b_D \in \mathcal{R}$ are parameters of the Decoder shared across all positions $i$ and $\hat{y}_i \in \mathcal{R}$ is the distance prediction at position $i$. The distance prediction at all position are reshaped into a matrix to get the final prediction $\hat{y} \in \mathcal{R}^{M,M}$. The entire model is trained using pairs of input $x$ and output $y$ datapoints with mean-squared error as the loss function.

## 3.2 End-to-end Mapping and Planning

The SPT model described above is designed to predict action distances given a map as input. However, in many applications, the map of the environment is often not known. In such cases, an autonomous agent working in a realistic environment needs to predict the map from raw sensory observations. While it is possible to train a separate mapper model to predict maps from observations, this often requires map annotations which are expensive to obtain and often inaccurate. In contrast, demonstration trajectories consisting of observations and optimal actions are more readily available or easier to obtain in many robotics applications. One of the key benefits of learning-based differentiable spatial planning is that it can be used to learn mapping just from action supervision in an end-to-end fashion without having access to ground-truth maps. To demonstrate this benefit, we train an end-to-end mapping and planning model to predict action distances from sensor observations for both navigation and manipulation tasks.

The end-to-end mapping and planning model consists of two modules, a Mapper ($f_M$) and a Planner ($f_P$), as illustrated in Figure 4. The Mapper is used to predict the map $\hat{m}$ from sensor observations $o$ and the Planner is a spatial planning model to predict action distances, $\hat{y}$, from the predicted map $\hat{m}$:

$$\hat{y} = f_P(\hat{m}) = f_P(f_M(o))$$

For navigation, $o$ is the set of first-person RGB camera images each of size $3 \times H \times W$. We sample 4 images, one for each orientation, at each valid location in the map. For invalid locations, we pass an empty image of 0s for all orientations. Thus, for a map of size $M \times M$, observation $o$ consist of $4M^2$ images for all locations and 4 orientations similar to the setup in Lee et al. (2018). For manipulation, $o$ is a top-down view of the operational space with obstacles of size $P \times P$, where each element is 1 or 0 denoting obstacles or free space. We use different Mapper architectures for navigation and manipulation experiments.

The Navigation Mapper module predicts a single value between 0 and 1 for each image in $o$ indicating whether the cell in the front of the image is an obstacle or not. The architecture of the Navigation mapper consists of ResNet18 convolutional layers followed by fully-connected layers (see Appendix for details). Each cell can have upto 4 predictions (from images corresponding to the four neighboring cells facing the current cell), which are aggregated using max pooling to get a single prediction. This map prediction consisting of continuous values is passed to the Planner module.

The Manipulation Mapper module needs to predict which configurations of the arm would lead to a collision. To predict whether a particular configuration $(\theta_1, \theta_2)$, the mapper module needs to check whether any point in this configuration consists of an obstacle. A Transformer-based model is well suited to learn this function as well as it can attend to arbitrary locations in the operational space to predict the obstacles in the configuration space. We use the same architecture of the SPT model as the Manipulation Mapper as well, with the only difference being the encoder consisting of $3 \times 3$ kernel size convolutional layers instead of $1 \times 1$ to encode the $P \times P$ observation space to a $M \times M$ representation.

The Planner module is the entire SPT model with encoder, transformer and decoder units as described in the previous subsection. It is pretrained on synthetic maps and its weights are frozen during end-to-end training. We train the entire end-to-end Mapping and Planning model with pairs of input observations $o$ and output action distances $y$ using standard supervised learning with the mean-squared error as the loss function: $\mathcal{L} = MSE(y, \hat{y}) = 1/|L| \sum_{i \in L} (y_i - \hat{y}_i)^2$, where $L$ is the set of navigable locations. Since the planning module is pretrained and it expects a structured map input, the mapper model needs to predict the map accurately such that the predicted map, when passed through the planner, minimizes the action level loss.

## 4 EXPERIMENTS & RESULTS: SPATIAL PLANNING

**Datasets.** We generate synthetic datasets for training the spatial planning models for both navigation and manipulation settings. For the navigation setting, we perform experiments with $M \times M$ maps with three different map sizes, $M \in \{15, 30, 50\}$. For manipulation, we experiment with two map sizes, $M \in \{18, 36\}$, corresponding to $20°$ and $10°$ bins for each link. In each map, we randomly generate $o_{min} = 0$ to $o_{max} = 5$ obstacles. Dataset generation details are provided in the Appendix.

For both the settings, we generate training, validation, and test sets of size $100K/5K/5K$ maps. The set of maps in each set are distinct. For each map, we choose a random free space cell as the goal location. The action space consists of 4 actions: north, south, east, west. For the navigation task, the map boundaries are considered as obstacles, while for the manipulation task the cells on the left and right boundaries and top and bottom boundaries are connected to each other since angles are circular. The ground truth shortest path distances are calculated using the Dijkstra algorithm (Dijkstra et al., 1959). Unreachable locations and obstacles are denoted by $-1$ in the ground truth.

In addition to testing on unseen maps with the same distribution, we also test the spatial planning models on two types of out-of-distribution datasets: **More Obstacles** where we generate $o_{min} = 15$ to $o_{max} = 20$ obstacles per map, and **Real-world** where the top-down maps are generated from reconstructions of real-world scenes in Gibson dataset (Xia et al., 2018).

**Hyperparameters and Training.** For training the SPT model, we use Stochastic Gradient Descent (Bottou, 2010) for optimization with a starting learning rate of 1.0 and a learning rate decay of 0.9 per epoch. We train the model for 40 epochs with a batch size of 20. We use $N = 5$ Transformer layers each with $h = 8$ attention heads and a embedding size of $d = 64$. The inner dimension of the fully connected layers in the transformer is $d_{fc} = 512$. We use the same architecture with the same hyperparameters for training the SPT model for both navigation and manipulation for all map sizes. We will open-source the code including dataset generation, model implementation and training.

| Method | Navigation | | | Manipulation | | Overall |
|--------|------|------|------|------|------|---------|
| | M=15 | M=30 | M=50 | M=18 | M=36 | |
| VIN | 86.19 | 83.62 | 80.84 | 75.06 | 74.27 | 80.00 |
| GPPN | 97.10 | 96.17 | 91.97 | 89.06 | 87.23 | 92.31 |
| **SPT** | **99.07** | **99.56** | **99.42** | **99.24** | **99.78** | **99.41** |

Table 1: **Generalization to in-distribution maps.** Table showing the average planning accuracy of the proposed model Spatial Planning Transformer (SPT) as compared to the baselines on in-distribution test sets for both the navigation and manipulation experiments.

| Method | Navigation | | | | | | Manipulation | | Overall |
|--------|------|------|------|------|------|------|------|------|---------|
| | More Obstacles | | | Real-World | | | More Obstacles | | |
| | M=15 | M=30 | M=50 | M=15 | M=30 | M=50 | M=18 | M=36 | |
| VIN | 49.05 | 62.05 | 70.64 | 49.91 | 56.67 | 71.16 | 65.27 | 59.81 | 60.57 |
| GPPN | 90.68 | 89.93 | 84.86 | 90.11 | 91.07 | 88.32 | 79.86 | 80.79 | 86.95 |
| **SPT** | **93.34** | **92.71** | **92.03** | **95.96** | **94.70** | **95.39** | **98.16** | **99.18** | **95.18** |

Table 2: **Generalization to out-of-distribution maps.** Table showing the average planning accuracy of the proposed model Spatial Planning Transformer (SPT) as compared to the baselines on out-of-distribution test sets for both the navigation and manipulation experiments.

**Baselines.** We use prior spatial planning models as baselines. These include Value Iteration Networks (VIN) (Tamar et al., 2016) and Gated Path-Planning Networks (GPPN) (Lee et al., 2018). For tuning the hyperparameter ($K$) for the number of iterations in both the baselines, we consider all values of $K$ in multiples of 10 such that the inference time of the baseline is comparable to the inference time of the SPT model ($\leq 1.1$ times). For each setting, we tune $K$ and the learning rate to maximize performance on the validation set.

**Metrics.** We use average action prediction accuracy as the metric. Distance prediction is converted to actions by finding the minimum distance cell among the 4 neighboring cells for each location. When multiple actions are optimal, predicting any optimal action is considered to be a correct prediction. The accuracy is averaged over all free space locations over all maps in the test set.

**Results.** The planning accuracy of all the methods for both the navigation and manipulation tasks on the in-distribution test sets are shown in Table 1 and on the out-of-distribution test sets are shown in Table 2. The proposed SPT model outperforms both the baselines across all settings achieving an overall accuracy of 99.41% vs 92.31% (in-distribution) and 95.18% vs 86.95% (out-of-distribution) as compared to the best baseline. The performance of the SPT model is stable as the map size increases while the performance of the baselines drop considerably. We believe this is because both the baselines need to use a larger number of iterations to cover a larger map ($K = 60$ iterations for GPPN and $K = 90$ iterations for VIN for $M = 50$) since the information propagation is local in VIN and GPPN. The optimization becomes difficult for such deep models. In contrast, the SPT model uses a constant $N = 5$ layers for all map sizes.

The improvement in the performance of SPT over the baselines is larger in the manipulation task because the baselines based on convolution operations are not well suited for propagating information looping over the edges of the map. In contrast, the SPT model can use self-attention to attend to any part of the map and learn to propagate information over the map edges.

**Visualizations.** In Figure 5, we show examples of predictions of the SPT model as compared to the baselines for 3 different input maps and goals from 3 different test sets. The examples show that the baselines are not able to predict the distances of distant cells accurately. This is because they propagate information in a local neighborhood that can not reach distant cells in the limited inference time budget ($K = 30$ for VIN and $K = 20$ for GPPN). In contrast, the SPT is able to predict distances of distant cells more accurately with $N = 5$ layers indicating that it learns long-range information propagation. Additional examples are provided in the Appendix.

| | Runtime per map in ms | | |
|--------|------|------|------|
| Method | M=15 | M=30 | M=50 |
| Dijkstra | 4.17 | 43.82 | 371.05 |
| A* | 3.02 | 35.38 | 294.70 |
| SPT | 2.44 | 4.72 | 18.35 |

Table 3: **Runtime comparison.** Comparison of average runtime per map in milli seconds for different methods. All values are averaged over 10000 maps.

**Runtime Comparison.** To demonstrate one of the benefits of learning-based planners over classical planning algorithms, we compare the runtime of SPT to Dijkstra (Dijkstra et al., 1959) and A* (Hart

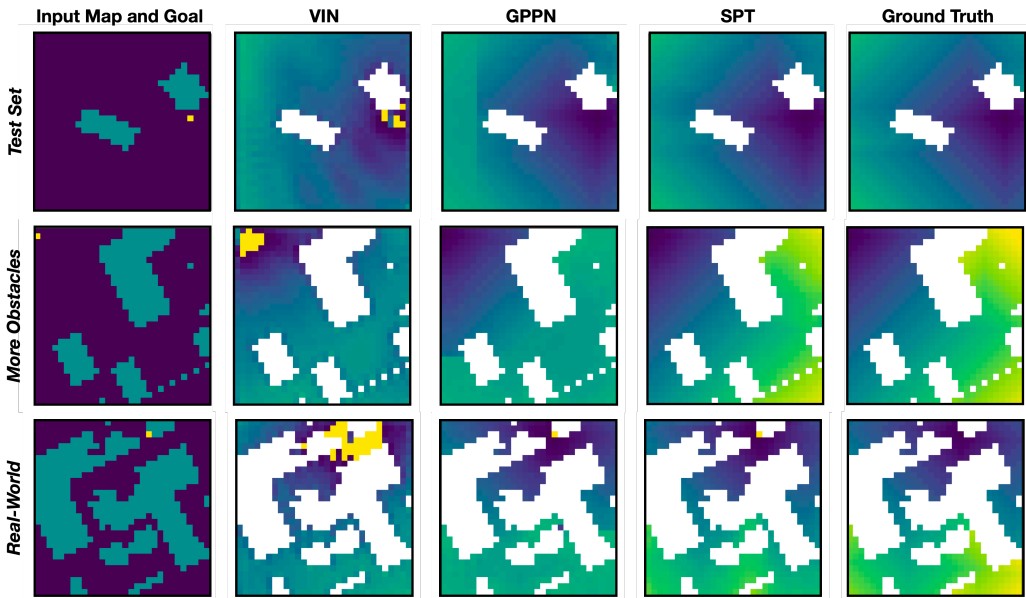

**Figure 5: Spatial Planning Examples.** Figure showing 3 examples of the input, the predictions using the proposed SPT model and the baselines, and the ground truth for map size $M = 30$. The obstacles are shown in blue, free space in purple and goal in yellow in the leftmost input column. The predictions and ground truth in the rest of the column are color-coded from blue to yellow to represent increasing action distance.

et al., 1968) algorithms in Table 3. The results indicate that SPT is $1.24\times$ to $20.22\times$ faster than classical planning algorithms with the runtime benefit improving with the increase in map size.

## 5 EXPERIMENTS & RESULTS: END-TO-END MAPPING AND PLANNING

In the above experiments, we compared the planning performance of different methods under perfect knowledge of the map $m$. In this section, we test the efficacy of spatial planning methods when map $m$ is unknown and needs to be predicted from sensor observations $o$.

**Datasets.** For manipulation, we generate synthetic datasets of size $M = 18$ using the same process as described in Section 4. We discretize the operation space into a $P \times P$ image with $P = 90$ which is used as the observation $o$. The train/test sets are of size 100K/5K.

For navigation, we use the Gibson dataset (Xia et al., 2018) to sample maps of size $M = 15$ where each cell is $0.25m^2$ area. We get the camera images at the navigable locations in all 4 orientations using the Habitat simulator (Savva et al., 2019). The set of camera images each of size $3 \times H \times W$ act as the observation $o$ for the navigation task, where $H = W = 128$. The train and test sets consist of 72 and 14 distinct scenes identical to the standard train and val splits in the Habitat simulator. We sample 500 maps in each scene creating training/test sets of size 36K/7K. Each sampled map is rotated to a random orientation.

**Training.** We load the weights of different models trained on synthetic data from the previous section. We then train the end-to-end model using the same action distance prediction loss while keeping the planner weights frozen. The architecture of the mapper module is identical across different planning methods. **Metrics.** We report both map accuracy and planning accuracy for both the tasks.

**Baselines.** In addition to using VIN and GPPN as baselines, we also use a classical mapping and planning baseline for the navigation task. Since there is no depth input available, we used Monocular depth estimation model from Hu et al. (2019) for predicting the map which is then used for planning using Dijkstra as suggested by Mishkin et al. (2019).

**Results.** Table 4 shows the end-to-end mapping and planning results. SPT outperforms both GPPN and VIN by a large margin across both the tasks achieving an overall plan accuracy of 82.29% vs 63.91%. Table 4 also shows that the mapper learnt using end-to-end training with a pretrained SPT model is able to achieve an accuracy of 98.96% for manipulation and 82.58% for navigation, without receiving any map-level supervision. SPT also outperforms the classical mapping and planning baseline. These results demonstrate a key benefit of learning-based differentiable planners

| Method | Navigation | | Manipulation | | Overall | |
|---|---|---|---|---|---|---|
| | Map Acc | Plan Acc | Map Acc | Plan Acc | Map Acc | Plan Acc |
| Classical | 64.43 | 45.20 | - | - | - | - |
| VIN | 60.92 | 47.77 | 81.25 | 66.45 | 71.08 | 57.11 |
| GPPN | 69.06 | 45.70 | 85.57 | 82.13 | 77.31 | 63.91 |
| **SPT** | **82.58** | **66.16** | **98.96** | **98.42** | **90.77** | **82.29** |

**Table 4: End-to-End Mapping and Planning Results.** Table showing the average mapping and planning accuracy of the proposed model Spatial Planning Transformer (SPT) as compared to the baselines for end-to-end mapping and planning experiments.

as compared to classical analytical planning algorithms. As SPT outperforms VIN and GPPN at spatial planning, it also leads to a better map accuracy (90.77% vs 77.31%).

### 5.1 SPARSE AND NOISY SUPERVISION

In the above experiments, we assumed access to perfect and dense action-distance supervision. In practice, if we were to get supervision from human trajectories, the supersion could be *sparse*, as we might not have access to the optimal distance from all locations in the map, and *noisy* as humans might not take the optimal actions always and computing distances from human trajectories might be noisy. To study the effect of not having dense and perfect supervision for training the end-to-end mapping and planning model, we consider three settings:

**Noisy supervision:** We add zero mean gaussian noise to all ground-truth distance values with standard deviation, $\sigma = 1$.

**Sparse supervision:** Instead of providing distances from all navigable locations in the ground truth, we provide distances for only 5 trajectories to the same goal in the training maps.

**Noisy and Sparse supervision:** We provide noisy distances for only 5 trajectories as supervision.

Figure 6 shows an example of noisy and sparse supervision. The results are shown in Table 5. The SPT model maintains performance benefits over the baselines under all the settings. Interestingly, under sparse supervision, the map prediction accuracy drops, but the planning accuracy does not drop as much. This is because the model learns to predict minimum map required to predict the action distances of all valid locations accurately as seen in examples shown in Figure 15 in the Appendix.

## 6 RELATED WORK

Path planning in known or inferred maps, also known as motion planning in robotics, is a well explored problem led by the seminal papers (Canny, 1988; Kavraki et al., 1996; LaValle & Kuffner Jr, 2001; Karaman & Frazzoli, 2011). Although there are learned variants of motion planners proposed in the literature using gaussian processes (Ijspeert et al., 2013; Ratliff et al., 2018), data-driven motion planners using neural networks is a recent direction (Qureshi et al., 2019; Bhardwaj et al., 2020; Qureshi et al., 2020). Prior work has also studied the use of neural networks to learn the heuristics and sampling stratergies in classical planners Ichter et al. (2018); Guez et al. (2018); Satorras & Welling (2020); Khan et al. (2020). Learning for planning is more common in Markov Decision Process (MDPs) for computing value function via dynamic programming based value iterations (Bellman, 1966; Bertsekas et al., 1995). Planning and learning in neural networks has been explored (Ilin et al., 2007) with a successful general formulation provided by value iteration networks (VIN) (Tamar et al., 2016) with follow-ups to improve scalability and efficiency (Lee et al., 2018; Karkus et al., 2017; Nardelli et al., 2018; Schleich et al., 2019; Khan et al., 2018; Chen et al., 2020). However, these models only capture local value propagation using CNNs and are mostly applied in navigation setups. In contrast, proposed SPTs capture long-range spatial dependency and easily scale to both navigation and manipulation.

Differentiable planning structure has also been explored in reinforcement learning with model-free methods (Silver et al., 2017; Oh et al., 2017; Zhu et al., 2017; Farquhar et al., 2018) as well as off-policy RL (Eysenbach et al., 2019; Laskin et al., 2020). Recent works also backpropagate through learned planners to train the policy (Pathak et al., 2018; Srinivas et al., 2018; Amos et al., 2018) and use imagined rollouts of a learned world model for long-term plans (Racanière et al., 2017; Hafner et al., 2019; Sekar et al., 2019). Unlike our work, these works lack the structure of a spatial planner.

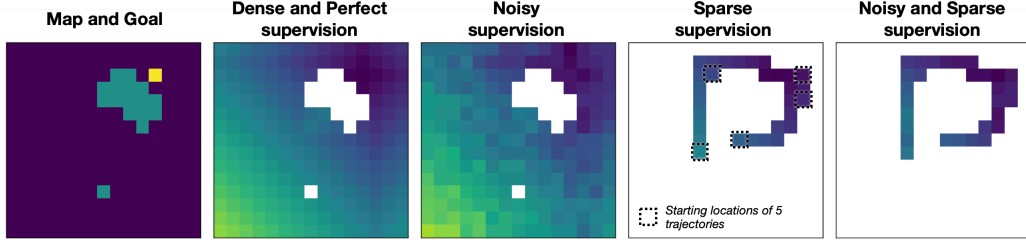

**Figure 6: Sparse and Noisy Supervision.** Figure showing examples of a map and goal with different levels of supervision. Noisy supervision adds gaussian noise to the ground truth distance values, and sparse supervision samples 5 trajectories for random starting locations to the goal location.

| Method | Dense and Perfect supervision | | Noisy supervision | | Sparse supervision | | Noisy and Sparse supervision | |
|---|---|---|---|---|---|---|---|---|
| | Map Acc | Plan Acc | Map Acc | Plan Acc | Map Acc | Plan Acc | Map Acc | Plan Acc |
| VIN | 81.25 | 66.45 | 75.68 | 60.78 | 70.16 | 60.23 | 70.22 | 58.97 |
| GPPN | 85.57 | 82.13 | 80.13 | 76.11 | 72.73 | 75.13 | 70.08 | 72.85 |
| **SPT** | **98.96** | **98.42** | **96.35** | **95.83** | **80.15** | **97.18** | **77.17** | **94.34** |

**Table 5: Sparse and Noisy Supervision Results.** Table showing the average mapping and planning accuracy of the proposed model Spatial Planning Transformer (SPT) as compared to the baselines for end-to-end mapping and planning experiments under noisy and sparse supervision settings for the manipulation task.

Decomposing learning a controller into mapping and planning is common in robot navigation (Khatib, 1986; Elfes, 1987). Some works have explored joint mapping and planning (Elfes, 1989; Fraundorfer et al., 2012). Maps can also be built from vision (Konolige et al., 2010; Fuentes-Pacheco et al., 2015) with a learned mapper (Parisotto & Salakhutdinov, 2017; Karkus et al., 2020). There has been some work on learning maps without using map annotations as well (Gregor et al., 2019). For navigation specific applications, recent works proposed joint mapping and planning for navigation (Gupta et al., 2017; Zhang et al., 2017; Savinov et al., 2018; Chaplot et al., 2020). However, most of these works either require access to ground truth map or assume interaction. Hence, they will first need to be trained in simulation. In contrast, we show results when the map is not known to the agent by learning just from trajectories and can be directly learned from data collected in the real-world.

## 7 DISCUSSION

The SPT model is designed to learn long-range spatial planning and it outperforms baselines across multiple experimental settings on both navigation and manipulation tasks. End-to-end learning experiments demonstrate that the SPT model can deal with unknown maps by learning mapping without any map-supervision. However, there are some limitations that need to be addressed before learning-based planning methods can be used for large-scale applications. We showed that the SPT model scales much better with increasing map sizes as compared to the baselines, however larger map sizes lead to a high inference time. In the future, we plan to tackle larger map sizes by learning an encoding which reduces the size of the map before spatial planning. The action space was limited to 4 actions in our experiments. Actions can be more fine-grained with smaller cells in the map, which can also be incorporated by tackling larger map sizes in the future.

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

## A    BACKGROUND: TRANSFORMERS

The proposed spatial planning method is based on the Transformer model (Vaswani et al., 2017). A Transformer layer, denoted by $f_{\text{TL}}$, takes a tensor $X \in \mathcal{R}^{d \times S}$ as input, where $d$ is the embedding size and $S$ is the size of the input. It consists of two sublayers, a multi-head self-attention layer ($f_{\text{SA}}$) and a position-wise fully connected layer ($f_{\text{FC}}$). There is a residual connection around each sublayer, followed by layer normalization (Ba et al., 2016) (LN):

$$R = \text{LN}(f_{\text{SA}}(X) + X), \quad Y = f_{\text{TL}}(X) = \text{LN}(f_{\text{FC}}(R) + R)$$

where $R, Y \in \mathcal{R}^{d \times S}$ are the intermediate and final representations, respectively.

The multi-head self-attention ($f_{\text{SA}}$) layer has $h$ attention heads, each computes a scaled dot-product attention over queries $Q$, keys $K$ and values $V$, which are all different projections of the input $X$:

$$Q_i = W_{Q,i}^T X, \quad K_i = W_{K,i}^T X, \quad V = W_{V,i}^T X$$

$$Z_i = \text{Attention}(Q_i, K_i, V_i) = \text{softmax}\left(\frac{Q_i K_i^T}{\sqrt{d_k}}\right) V_i$$

where $Q, K \in \mathcal{R}^{d_k \times S}$, $V \in \mathcal{R}^{d_v \times S}$, $i \in 1, 2, \ldots, h$ $d_k$ and $d_v$ are hyper-parameters and all $W$s are parameters. The output of all attention heads, $Z_i$s, are concatenated and projected to the same dimension as the input. Finally, the position-wise fully connected ($f_{\text{FC}}$) layer applies two linear transformations to each position with a ReLU activations to the output of the multi-head attention.

## B    DATASET DETAILS

We generate synthetic datasets for training the spatial planning models for both navigation and manipulation settings. For the navigation setting, we perform experiments with $M \times M$ maps with two different map sizes, $M \in \{15, 30\}$. We randomly generate $o_{min} = 0$ to $o_{max} = 5$ obstacles in each map, where each obstacle is an rectangle at a random location with each side being a random length from 1 to $M/2$. All the rectangular obstacles are rotated in two random orientations.

For the manipulation setting, we consider a reacher task using a planar arm with 2 degrees of freedom. We use an operational space of size $P \times P$. Each link of the arm is of size $P/4$. The arm is centered at the center of the operational space. Let the orientation of two links be denoted by $\theta_1$ and $\theta_2$. We assume both the links can freely rotate in a plane, $\theta_1, \theta_2 \in [0, 2\pi)$. For each environment, we generate $o_{min} = 0$ to $o_{max} = 5$ circular obstacles centered at a random location $0.25P$ to $0.75P$ distance away from the center, with a random radius between $0.05P$ and $D - 0.15P$ where $D$ is the distance of the center of the obstacle from the center of the operational space. We convert each environment to a configuration space map of size $M \times M$, where each cell $(i, j)$ denotes whether the arm will collide with an obstacle when $\theta_1 = 2\pi i/M$ and $\theta_2 = 2\pi j/M$. We experiment with two map sizes, $M \in \{18, 36\}$, corresponding to $20°$ and $10°$ bins for each link. The choice of $P$ does not affect the map as the collision check for each cell in the configuration space is performed in the continuous operational space where all distances are relative to $P$.

## C    NAVIGATION MAPPER ARCHITECTURE DETAILS

The Navigation Mapper module predicts a single value between 0 and 1 for each image in $o$ indicating whether the cell in the front of the image is an obstacle or not. The architecture of the Navigation mapper consists of ResNet18 convolutional layers followed by 3 fully-connected layers of size 256, 128 and 1 as shown in Figure 7. Each cell can have upto 4 predictions (from images corresponding to the four neighboring cells facing the current cell), which are aggregated using max pooling to get a single prediction.

## D    HIGHER DIMENSIONAL STATE AND ACTION SPACES

To test whether SPT maintains performance benefits in higher dimensional state and action spaces, we conducted some experiments for the navigation task. We relaxed the action space from 4 actions

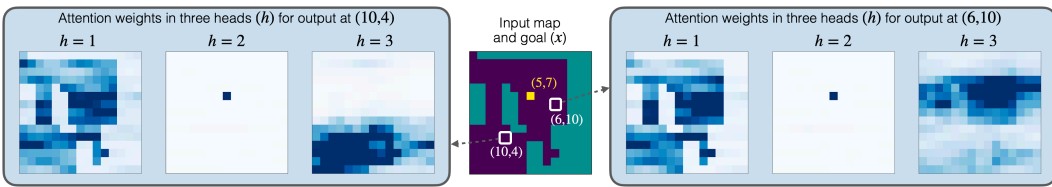

**Figure 7: Navigation Mapper Architecture.** Figure showing the architecture of the Navigation Mapper.

to 100 actions by just allowing the agent to take any action in a 10x10 grid around it (and using a low-level controller to go to any cell). The state space used for planning is discretized but the agent moves in a continuous state space in the Habitat simulator. We compute the continuous ground truth distance using Fast Marching Method (instead of Dijkstra) for training with a larger action space, which allows us to accurately compute distance for all locations and not be constrained by axis-aligned actions and distances.

SPT and the baselines are trained only on synthetic navigation mazes for this experiment. During evaluation, we assume perfect map based on part of the environment seen in the observations so far for planning. If the overall map size at this level of discretization is higher than planning map size, we simply use greedy planning in a window around the agent resulting in an "anytime" variant just like the classical RRT algorithms. This setup results in much finer-grained action space as shown in the demo example here [1]. SPT achieve navigation success rate of 78.0% as compared to 47.2% for GPPN and 43.5% for VIN baselines.

## E    ATTENTION VISUALIZATION

We show the visualization of attention map corresponding to two different locations in Figure 8. Interestingly, we noticed three consistent patterns: a) at least one of the attention head out of eight captures obstacles (left), b) one of the attention heads focuses on goal location (middle), and c) some attention maps focus on nearby obstacles to get accurate planning distance (right).

**Figure 8: Attention Visualization.** Visualization of the attention heads learned by Spatial Planning Transformers. SPTs learn an attention for each location in the map with respect to every other location.

## F    EXAMPLES

We show additional examples for navigation task for in-distribution test set (in Figure 9), out-of-distribution More Obstacles test set (in Figure 10) and Real-World test set (in Figure 11) each with map size $M = 30$. Additional examples for manipulation task are shown for in-distribution test set (in Figure 12) and for out-of-distribution More Obstacles test set (in Figure 13).

We also visualize examples for the end-to-end mapping and planning experiments for the manipulation task. We show examples of map and action distance predictions using the SPT model trained with dense and perfect supervision in Figure 14 and with noisy and sparse supervision in Figure 15.

---

[1]https://youtu.be/sNHDhb3t7AM

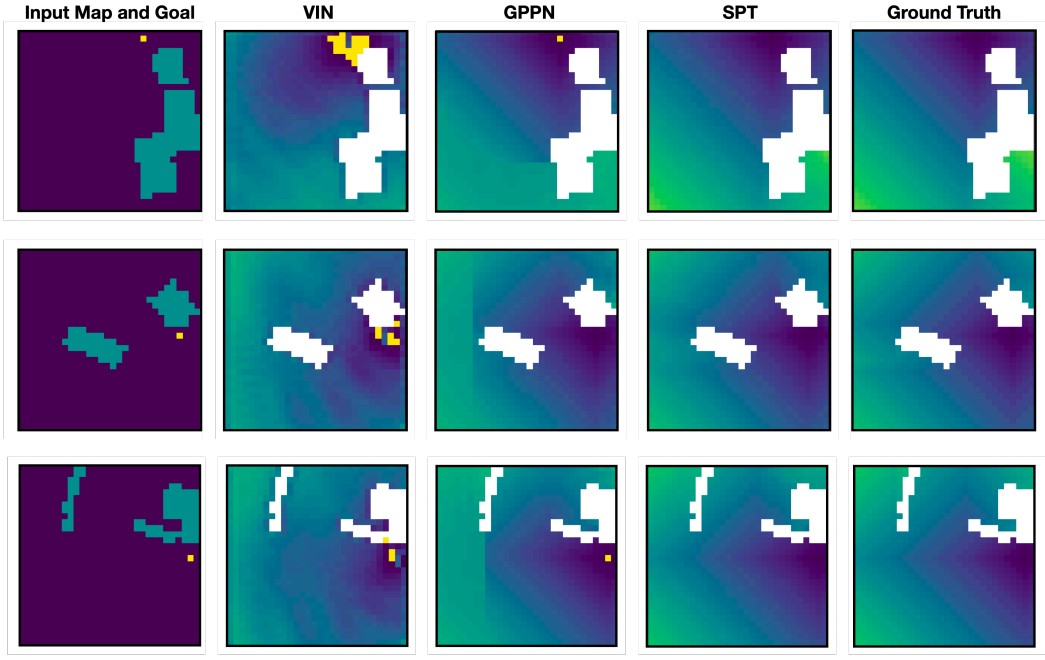

**Figure 9: Navigation in-distribution test set examples.** Figure showing 3 examples of the input, the predictions using the proposed SPT model and the baselines, and the ground truth for the Navigation in-distribution test set for map size $M = 30$.

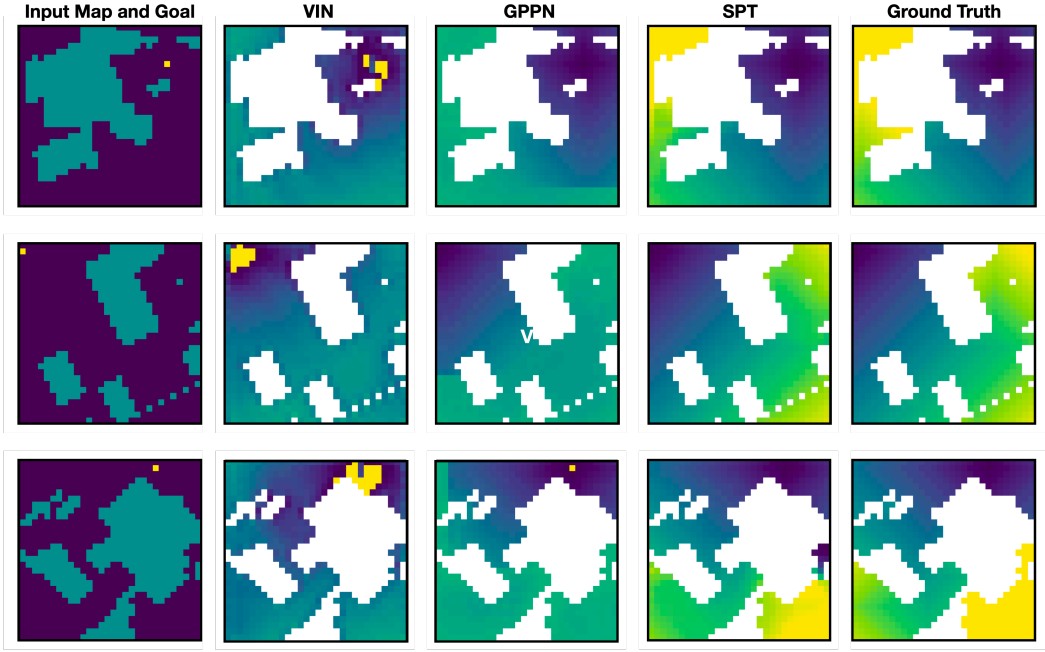

**Figure 10: Navigation out-of-distribution More Obstacles test set examples.** Figure showing 3 examples of the input, the predictions using the proposed SPT model and the baselines, and the ground truth for the Navigation out-of-distribution More Obstacles test set for map size $M = 30$.

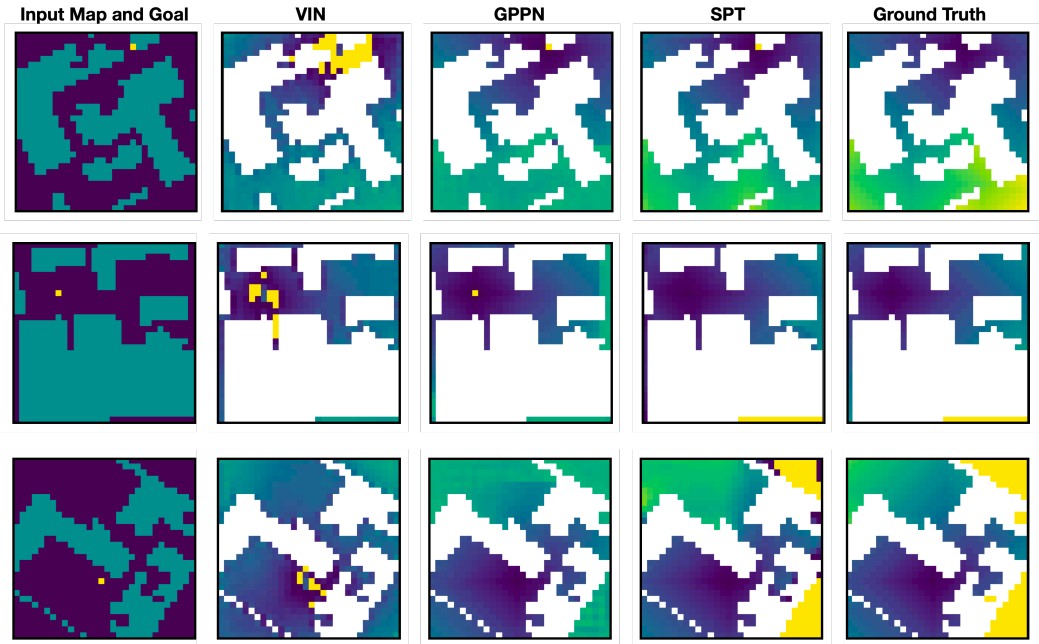

**Figure 11: Navigation out-of-distribution Real-World test set examples.** Figure showing 3 examples of the input, the predictions using the proposed SPT model and the baselines, and the ground truth for the Navigation out-of-distribution Real-World test set for map size $M = 30$.

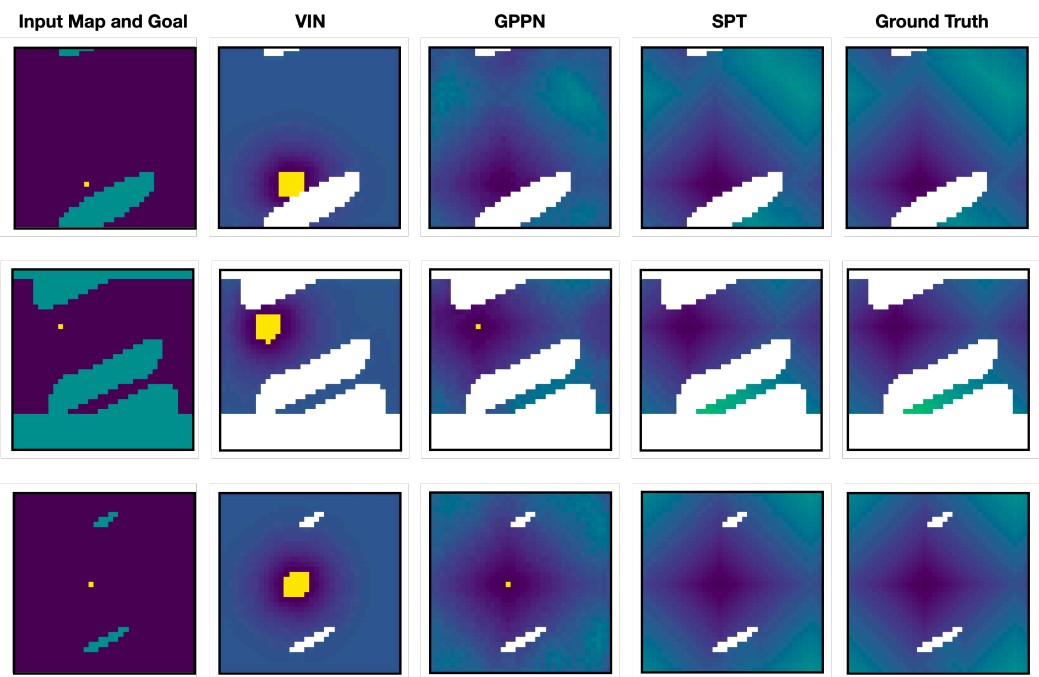

**Figure 12: Manipulation in-distribution test set examples.** Figure showing 3 examples of the input, the predictions using the proposed SPT model and the baselines, and the ground truth for the Manipulation in-distribution test set for map size $M = 36$.

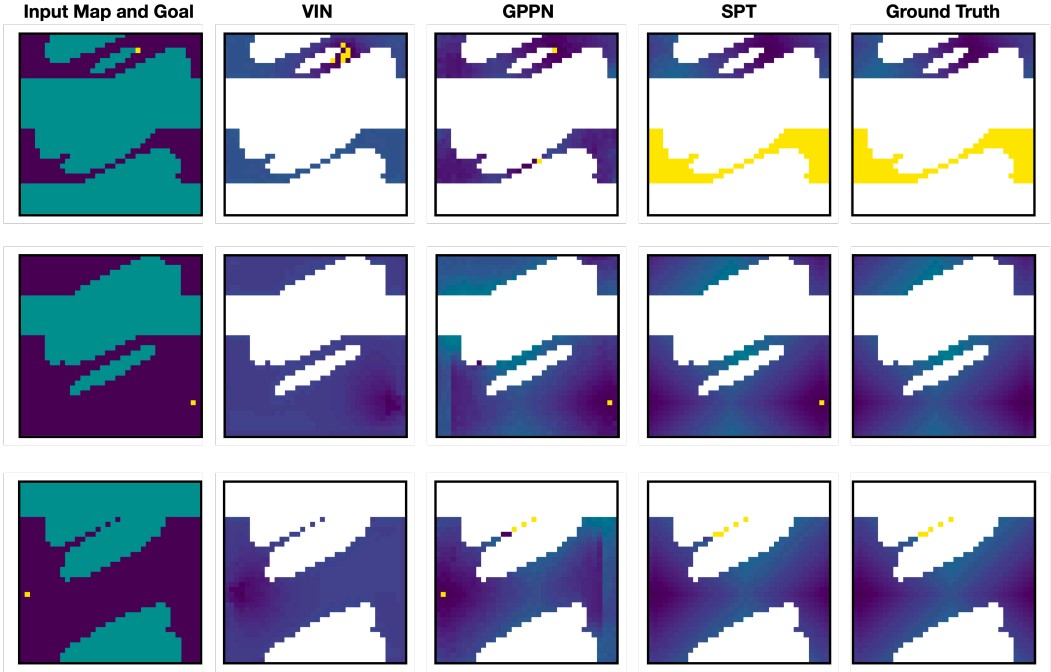

**Figure 13: Manipulation out-of-distribution More Obstacles test set examples.** Figure showing 3 examples of the input, the predictions using the proposed SPT model and the baselines, and the ground truth for the Manipulation out-of-distribution More Obstacles test set for map size $M = 36$.

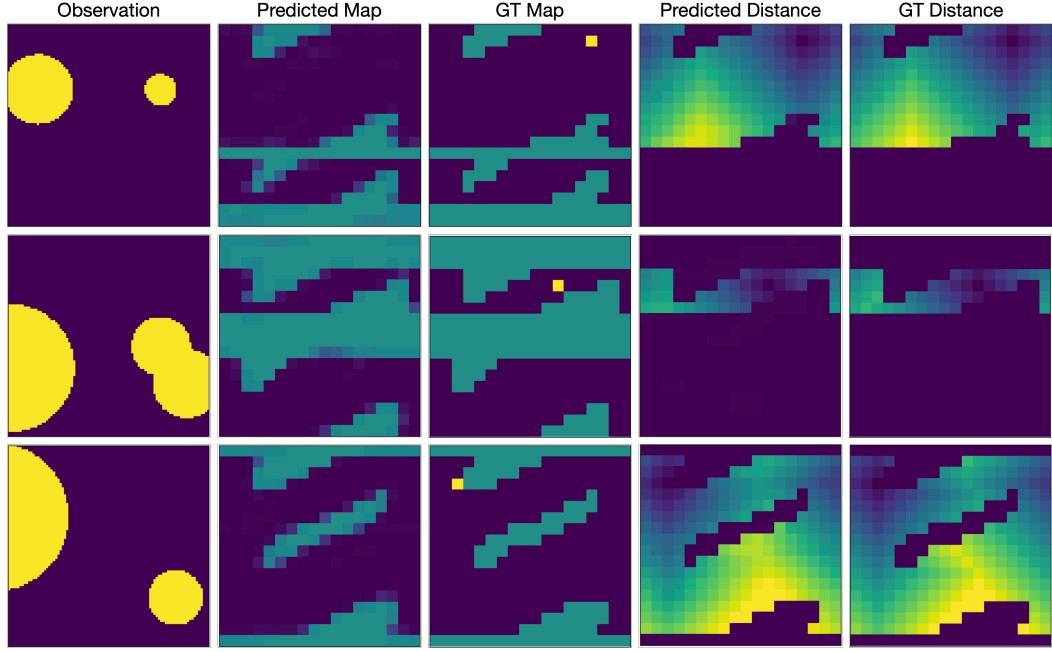

**Figure 14: Dense and Perfect Supervision.** Figure showing examples of map and distance predictions using the SPT model trained with dense and perfect action-level supervision.

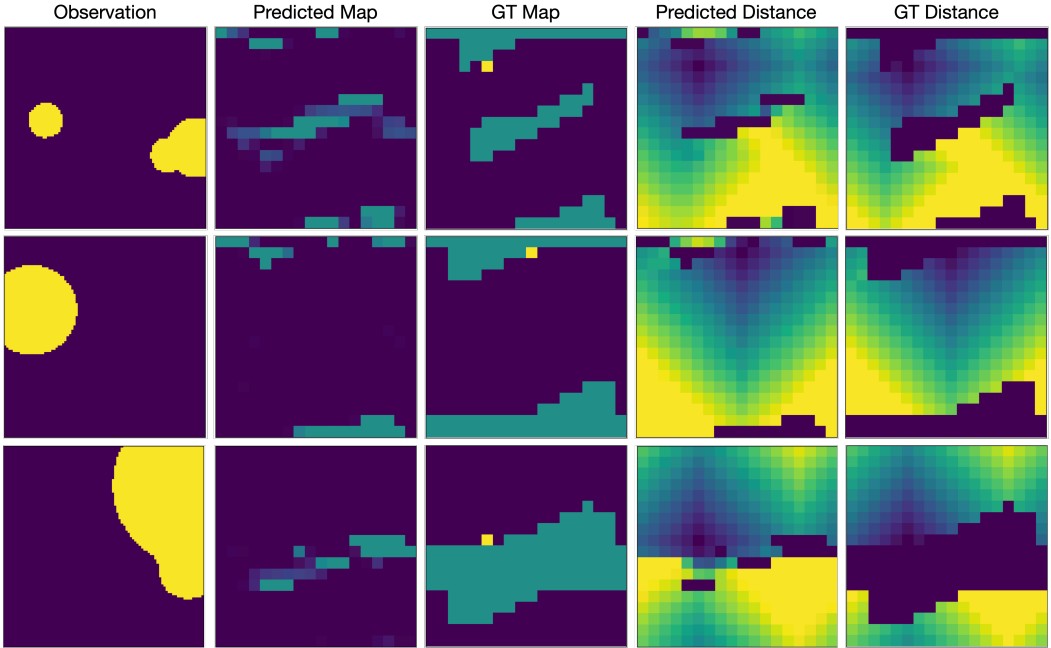

**Figure 15: Sparse and Noisy Supervision.** Figure showing examples of map and distance predictions using the SPT model trained with sparse and noisy action-level supervision.

