# OpenReview forum: "Differentiable Spatial Planning using Transformers"
_ICLR.cc/2021/Conference — Reject_

### Official Review · AnonReviewer3 · 2020-10-25
**Transformer for planning: interesting, but few points need clarification.**

**Rating:** 6
**Confidence:** 3

**Review:**


Summary:

The paper provides an interesting direction in the field of spatial path planning. The method is interesting as it is fully learnt in an end to end fashion. The key idea is to use a transformer like architecture to model long range dependencies. Also the paper extend its findings to out of distribution maps and  the cases where the ground truth map is not known to the agent.

##########################################################################

Reasons for score:

Overall, I think the paper is slightly below the acceptance threshold. The motivations and the architecture are clearly presented, but I think there is a lack of clarity in the way the methods and the results are presented, especially in the most interesting section, the one about the mapper. Also the analysis are limited to reconstructions of the map and some attention weights in the appendix.

##########################################################################

Pros:

1. Interesting new approach with potential applications in both navigations and manipulations task.
2. The out of distribution results are convincing.

Cons:

1.  Although the method is a clear advantage over previous ones it still requires to know the ground truth distance to get the map.

2. Instead of using a paragraph to explain the transformer architecture, which is relatively well know, my suggestion would be clarifying the mapper section with more notation to clearly state the loss. This would already help to most likely increase my score.

3. In section 3.2 the authors claim that “While it is possible to train a separate mapper model to predict maps from observations, this requires map annotations which are expensive to obtain and often inaccurate”. I don’t think this is true: e.g. Gregor K. et al., 2019 ->  https://arxiv.org/pdf/1906.09237.pdf present an example of how a map can be learnt.

4. It is not clear how much the model relies on having perfect distances as supervision vs. noisy ones. An analysis on this topic I think would be important.

5. No Analysis of computational complexity

---

> ### Author Response · Authors · 2020-11-25
> **Response to AnonReviewer3**
>
> We thank the reviewer for helpful comments and suggestions to which we respond below:
>
> - 1. and 4. Regarding noisy supervision:
>
> Thanks for the great suggestion! It is true that in practice, the distance values in the ground-truth can be noisy. We have added some more experiments to test the performance of the method given noisy distance supervision in Section 5.1. The results are presented in Table 5 that show that SPT performs well even with noisy supervision, achieving 95.83% planning accuracy as compared to 76.11% for GPPN [2] and 60.78% for VIN [1].
>
> - 2. Regarding mapper details:
>
> Thanks for the suggestion, we have updated the description of the mapper section to include more details along with the loss function (see Section 3.2). We have also added the details of the Navigation mapper architecture to the Appendix.
>
> - 3. Regarding related work:
>
> Thanks for pointing out a relevant paper. We have revised the statement and added the paper to the related work.
>
> - 5.  Regarding computational complexity:
>
> The computational complexity of SPT is O(N\*M^2) where N is the number of transformer layers and M is the map size. The computational complexity of both the baselines VIN and GPPN is linear in the number of iterations O(K\*M^2) where K is the number of iterations and M is the map size. The runtime of the learning-based methods varies based on the choice of the hyperparameters. For all our experiments, we chose the value of K in baselines which led to comparable runtimes with N=5 transformer layers. We have also added a runtime comparison of SPT with the classical planning methods in Table 3.
>
> [1] Tamar et al. Value Iteration Networks. NeurIPS, 2016
>
> [2] Lee et al. Gated Path Planning Networks. ICML, 2018

---

### Official Review · AnonReviewer2 · 2020-10-27
**This paper aims to solve spatial planning using a neural network. The idea is to encode inductive biases about spatial environments into a neural planner by learning on a set of environments. A multi-head attention mechanism is used to propagate information across long distances.**

**Rating:** 7
**Confidence:** 3

**Review:**

The paper correctly identifies some of the shortcomings of classical planning methods. The idea to use a transformer for planning seems promising and that is backed up by experimental results where the method outperforms previous work. I think the paper has some interesting ideas and the experimental section seems promising, but some assumptions are not stated clearly enough (I will discuss specific points under "Cons"). I recommend accepting the paper on the basis that the ideas presented are relevant to the community.

### Pros

* To the best of my knowledge, this is the first paper to propose combining transformers with neural planning in the style of value iteration networks.
* The paper deals with both robotic manipulation and navigation in the same framework. It is nice to see these related problems being tackled jointly.
* The method has an explicit component for creating maps, which is generally omitted in the framework of value iteration networks and their derivatives.
* The experimental results clearly demonstrate an edge over other VIN-type approaches.

### Cons

* The section on the mapper makes it sound like it maps one observation to an entire map, though in almost all cases, one observation won't be enough to do so. No mechanism for aggregating observations over time is described. The only way I can see this working is if the method gets an exhaustive dictionary of images from many locations and orientations in the environment, such as the 3D experiments by [Lee et al.](https://arxiv.org/pdf/1806.06408.pdf) The experiments section also gives this impression. This does not invalidate the approach but I would like this to be stated more clearly and earlier on in the text.

* The end-to-end module is trained using action-distance labels. This is basically assuming we have access to shortest-distance labels or a full value function. I don't understand how this is preferable to assuming we have full knowledge of the obstacles, which the paper seems to imply by saying:

  > This essentially allows us to train a mapping module without any map-level supervision using a pretrained planner and action-level supervision.

  My issue here is that I don't understand how we could get exhaustive shortest-distance labels without having access to an occupancy map or a list of obstacles. It's fine to assume we have access to such things during training, I just want this to be stated clearly.

* Some metric that is a direct measure of navigation success would be beneficial (e.g. percentage of successful attempts or [success-path-length](https://arxiv.org/pdf/1807.06757.pdf)). The action-prediction-accuracy is a proxy to this but one can imagine situations where a model can still have high navigation success while taking sub-optimal actions some of the time.

---

> ### Author Response · Authors · 2020-11-25
> **Response to AnonReviewer2**
>
> We thank the reviewer for helpful comments and suggestions to which we respond below:
>
> - Regarding details of end-to-end mapping and planning experiments:
>
> The Navigation mapper receives images across the map as input. We revised the description of end-to-end mapping and planning to include more details and state the set of images given as input and the mechanisms used for aggregating obstacles explicitly (see Section 3.2).
>
> - Regarding dense labels:
>
> This is a great observation and suggestion! Exhaustive shortest-path distances would not be available without the map in practice. We have made revisions to clarify this. We also conducted additional experiments to test whether the method can perform well when given access only to distance to only a sparse subset of the locations. Instead of providing distances from all navigable locations in the ground truth, we provide distances for only 5 trajectories to the goal location in the training maps. The experiment is described in Section 5.1 and the results are presented in Table 5. SPT maintains performance benefits over the baselines even when given sparse supervision.
>
> - Regarding other metrics:
>
> We observed that the trend in success rate is very similar to the trend in the planning accuracy as also observed by Lee et al. [1]. We decided to report planning accuracy as it is a more stringent metric and it is more indicative of the planning performance. The success rate was often >99.9% for smaller maps which made it difficult for us to understand the difference between various methods and design choices.
>
> [1] Lee et al. Gated Path Planning Networks. ICML, 2018

---

### Official Review · AnonReviewer4 · 2020-11-02

**Rating:** 4
**Confidence:** 5

**Review:**

The paper proposes a method to infer goal distance maps for use in path planning using transformer networks. These 2D distance maps are evaluated on both planning and combined mapping and planning experiments on synthetic data.

While the proposed method seems technically sound and based on the experiments works as intended, it is not clear how the proposed system would be applicable in practice. Additionally, the motivations for the proposed approach over classical planning methods, such as run-time and incomplete environments, were not explored.

From a machine learning point of view, there is nothing particularly novel about the proposed method which uses well known and existing methods to build the planning framework. This is not a negative point for methods which develop interesting algorithms for applications. However, in the case of this paper, there are many questions regarding the applicability of the resulting system.
The proposed method produces a distance to goal map based on an obstacle representation. While such a distance map can be used with search methods to find a path, it does not constitute a path planning method itself. Furthermore, the proposed method is by design limited to 2D spaces. While this to some extent might be justifiable for ground-based robots, it precludes the use of the approach in challenging outdoor environments or on flying robots. More importantly, this reduces the applicability this method has in manipulation severely as the vast majority of manipulators used these days have six or seven degrees of freedom. These limitations are furthermore exaggerated by the very coarse environment discretization used in the examples of 0.25 cm or 10 degrees. While 0.25 cm grid sizes for navigation can be acceptable in large outdoor environments, the 10 degree discretization used would be unsuitable for most manipulation tasks.

One of the motivations given by the paper for the development of an end to end learning framework is the computational efficiency of having to only perform a single forward pass over the input. This, in theory, enables fast recomputation of a solution should the environment change. Given the 2D grid structure of the input and outputs, one would expect a comparison to widely used search methods such as A* and R*. These methods would be very fast to compute on the problem setups used in the experiments while being provably optimal. Providing a run-time comparison against such methods would have been good to see. Similarly, comparisons to RRT based methods, which in the limit are also provably optimal, would be appreciated.
While the combination of planning and mapping was interesting, it is not clear why this was needed, as methods capable of accurately mapping large scale environments exist without relying on learning systems. As the focus of the paper was on planning, this also caused some confusion on their relation.

Overall, while the proposed method works, it is not clear what practical benefit it has. As a deep learning approach, it requires a significant amount of training and has no guarantees to provide predictable results at all times. This is in stark contrast with decade-old methods such as A*, RRT*, and trajectory optimization which have such guarantees. The major limitation of the proposed method to 2D spaces severely limits its applicability with questions regarding scalability given the sizes used in the experimental section.


=== Post rebuttal ===

The inclusion of runtime comparisons with search based methods is a welcome addition, as it showcases that the method provides an interesting avenue to utilize GPUs for this kind of search task. Effectively trading up-front training time for deployment computational time. However, the question regarding the ability to handle larger state spaces is only touched on in the appendix and it is not clear from that description what the state space was. If the proposed approach works for planning in 6D environments or control of 7DoF manipulators then this is very interesting and should not be hidden away, so to speak, in the appendix but be highlighted in the main paper and showcased.

In light of the considerable amount of work that has been put into the revision I changed my score from 3 to 4. While I could see improvements and clarifications I could not see the main concern I had, handling of high dimensional problems, being addressed.

---

> ### Author Response · Authors · 2020-11-25
> **Response to AnonReviewer4**
>
> AnonReviewer4 has expressed their clear preference for classical planning based methods over all learning-based approaches. In doing so, their critique applies equally to all the past work in the learning community on differentiable planning methods and applications (for example [1,2,3,4,5]) and not specific to our particular paper. We should note that this is a flourishing subfield with publications in ICLR, NeurIPS, CVPR, etc. This includes various differentiable planning methods itself such as [1,2,3] and also many papers utilizing differentiable planners as a part of larger models (for example [4,5]). There are over a dozen papers in the last few years presenting different methods for learning to plan in general as discussed in the related work section.
>
> While it is difficult to rebut a purely ideological criticism, here are some arguments in favor of a learning-based approach to planning:
>
> - As discussed in the introduction, one benefit is that learning approaches can capture statistical regularities in data and can be efficient at inference as the plan is just a result of forward-pass through the network. We have added runtime comparisons with some classical methods in Table 3. The results indicate that the SPT is up to 20.22x times faster than classical planning algorithms with the runtime benefit improving with the increase in map size.
>
> - A learned solution can allow supervision from the action-level to be back-propagated through the planning to help improve mapping. They can not only provide the ability to deal with partial, noisy maps but also help build maps on the fly while acting in the environment with just action-level supervision without requiring access to map annotations. We conducted end-to-end mapping and planning experiments to highlight this benefit. We added classical mapping and planning as a baseline in this experiment. Since there is no depth input available, we used the Monocular depth estimation model from [7] for predicting the map which is then used for planning using Dijkstra as suggested by [6]. We chose monocular depth estimation as it performed better than stereo depth estimation and sparse 3D point cloud using ORBSLAM in experiments in [6] for similar environments. Classical mapping and planning baseline resulted in 45.20% planning performance as compared to 66.16% of SPT in the Habitat navigation test set. This is consistent with the results in [6] who also find that classical mapping and planning is not very effective given only RGB input.
>
> - A learning-based approach can also aid classical planning algorithms by learning a value function that can be used for sampling and search in classical planners instead of random sampling or heuristic-based search [8,9]. Our contribution is complementary to these works as SPT can be used as the value prediction module in these methods to improve sampling and search in classical planners.
>
> [1] Tamar et al. Value Iteration Networks. NeurIPS, 2016
>
> [2] Lee et al. Gated Path Planning Networks. ICML, 2018
>
> [3] Nardelli et al. Value Propagation Networks. ICLR, 2019
>
> [4] Gupta et al. Cognitive mapping and planning for visual navigation. CVPR 2017
>
> [5] Karkus et al. Qmdp-net: Deep learning for planning under partial observability. NeurIPS, 2017
>
> [6] Mishkin et al. Benchmarking Classic and Learned Navigation in Complex 3D Environments. arXiv preprint arXiv:1901.10915
>
> [7] Hu et al. Revisiting single image depth estimation: Toward higher resolution maps with accurate object boundaries. WACV, 2019
>
> [8] Ichter et al. Learning sampling distributions for robot motion planning. ICRA 2018.
>
> [9] Khan et al. Graph Neural Networks for Motion Planning. arXiv preprint arXiv:2006.06248

---

### Official Review · AnonReviewer1 · 2020-11-03
**Interesting idea; need more experiments**

**Rating:** 5
**Confidence:** 4

**Review:**


This paper presents Spatial Planning Transformers (SPTs); neural network modules that perform spatial planning over grid-like state spaces. The paper also goes on to present the idea that differentiable mapping and differntiable planning modules could be trained end-to-end, for better performance. This is evaluated against two baselines (value iteration networks (VINs) and generative path planning networks (GPPNs)) on small-scale navigation and manipulation tasks.


## Strengths

**S1** The central idea presented in this paper is quite interesting. Re-formulating the planning problem as learning an attention operator over an input map is both interesting, and challenging. The approach does have a strong motivation, considering the impact transformers have had in language processing. SPTs seek to replace memory-based models, such as LSTMs with purely attention-based models such as transformers, to potentially capture long-range dependencies.

**S2** The paper presents a promising idea in attempting to integrate differentiable mapping modules with differentiable planners. This could instigate further work on the lines of what kind of inductive biases are fruitful to have in neural planners.

**S3** The paper was very-well written and easy to follow.


## Weaknesses

I do find several aspects of the paper can be improved, and I list my most important observations here. I look forward to the author response, and am open to changing my evaluation if the concerns are adequately addressed.

**W1** Discrete action spaces: It seems like the proposed approach is tailored to discrete state-spaces or maps (and perhaps to discrete actions)? It could be interesting to know whether the approach scales to continuous state and/or action spaces. Upfront, it seems dubious, as it is unclear how the transformer blocks might apply to continuous maps.

**W2** Scalability issues: The assumption of a grid-like partitioning indicates that SPTs may exhibit unfavourable scaling properties? For instance, if one were to move from a 2D (planar) space to 3D, this seems like it will dramatically increase planning complexity. (While this can also be argued for "classical" planners like RRT, they often are accompanied by an exploration-exploitation tradeoff that lends them an anytime flavour)

**W3** Better placement wrt diff planning literature: I believe that SPTs can be better placed with respect to current literature on differentiable planning. For instance a few interesting approaches in this space include references [A-J]. Some of these [F, G, J] could be construed as concurrent work (but may be worth citing in a camera-ready version, for example). It would be a great addition to incorporate interesting new work in the robotics community on neural and/or differentiable planning [H, J]. It may also be worth using some of these approaches as baselines for evaluation.

**W4** More baselines: The current draft seems to only use VINs and GPPNs as baseline planning approaches. The presented method will seem better-placed in the context of other recent contributions if more baselines were to be introduced. For instance, universal planning networks, motion planning networks, semi-parameteric topological memory are all relevant baselines for navigation and/or manipulation.

**W5** In trying to present two important ideas (SPTs, diff mapping + planning), the current manuscript appears to lack a thorough study of neither. For instance, are there various combinations of differentiable mapping and planning strategies that can be evaluated? One would also have expected an evaluating against non-differentiable mapping + planning (i.e., "classical" robotics) strategies to strengthen this claim.


## Summary

In summary, while the paper presents a very interesting idea, there are several questions to be asked of the experimental demonstrations of the idea. This could benifit from a revision, and I strongly encourage the authors to address some of the issues raised above.


## References

[A] Learning to Plan in High Dimensions via Neural Exploration-Exploitation Trees - arXiv 2019

[B] Learning to search with MCTSnets - ICML 2018

[C] TreeQN and ATreeC: Differentiable Tree-Structured Models for Deep Reinforcement Learning - ICLR 2018

[D] Memory augmented control networks - ICLR 2018

[E] Learning to Plan in High Dimensions via Neural Exploration-Exploitation Trees - arXiv 2019

[F] Neural Enhanced Belief Propagation on Factor Graphs - arXiv 2020

[G] Neural Manipulation Planning on Constraint Manifolds - arXiv 2020

[H] Learning sampling distributions for robot motion planning - ICRA 2018

[I] Semi-parametric topological memory for navigation - ICLR 2018

[J] Graph neural networks for motion planning - arXiv 2020

---

> ### Author Response · Authors · 2020-11-25
> **Response to AnonReviewer1**
>
> We thank the reviewer for helpful comments and suggestions. We respond to all the comments mentioned in the review below:
>
> - Q1 & Q2: Regarding Discrete action spaces and scalability
>
> We believe all planning approaches involve some level of discretization of the state space, either explicitly in approaches like A* or Dijkstra, or implicitly by sampling nodes in the case of PRM, RRT*, etc. In practice, we would expect spatial planning to work at a higher level, resulting in discrete waypoints, and a low-level controller to move between waypoints dealing with continuous state and action spaces (for example in [1]).
>
> To test whether SPT maintains performance benefits in higher dimensional state and action spaces, we conducted some additional experiments for the navigation task. We relaxed the action space from 4 actions to 100 actions by just allowing the agent to take any action in a 10x10 grid around it (and using a low-level controller to go to any cell). The state-space used for planning is discretized to 25cm^2 cells but the agent moves in a continuous state space in the Habitat simulator. We compute the continuous ground truth distance using the Fast Marching Method (instead of Dijkstra) for training with a larger action space, which allows us to accurately compute the distance for all locations and not be constrained by axis-aligned actions and distances. If the overall map size at this level of discretization is higher than the planning map size, we simply use greedy planning in a window around the agent resulting in an “anytime” variant just like the classical RRT algorithms. This setup results in much finer-grained action space as shown in the demo example here: https://youtu.be/sNHDhb3t7AM. SPT achieve navigation success rate of 78.0% as compared to 47.2% for GPPN [3] and 43.5% for VIN [2]. This also highlights another benefit over VIN and GPPN as they are designed to work well only with axis-aligned action spaces such as NEWS (north, east, west, south). This experiment is added to Appendix D.
>
> There has been other work on scaling learning-based planners to high-dimensional continuous state space [9, 10] which include exploration-exploitation trade-offs (for example in [9] as you pointed out). Our contribution is complementary to these approaches as SPT can be used as the planning module in these models instead of CNN/LSTM based modules such as VIN and GPPN.
>
> There has been a lot of research on the scalability of transformers, for example via hashing of keys and query [4], using recurrence to capture variable lengths [5], using a low-rank approximation to linearize attention [6,7], etc. We used a vanilla transformer in our experiments, we believe incorporating these techniques can also allow us to scale SPT to much larger maps.
>
> - Q3: Regarding better placement wrt diff planning literature:
>
> Thanks a lot for pointing us to some very relevant papers! We have added discussion on comparison with these works in the related work.
>
> - Q4 and Q5: Regarding baselines:
>
> We considered using other learning-based baselines including some of the baselines suggested by the reviewer, however, we concluded that these baselines are not directly comparable as they involve graph-based planning on abstract latent states. This is very different from spatial planning as there’s no way for these baselines to leverage the spatial structure.
>
> As suggested by the reviewer, we added classical mapping and planning as a baseline in our end-to-end mapping and planning experiments for the navigation task. Since there is no depth input available, we used the Monocular depth estimation model from [11] for predicting the map which is then used for planning using Dijkstra as suggested by [8]. We chose monocular depth estimation as it performed better than stereo depth estimation and sparse 3D point cloud using ORBSLAM in experiments in [8] for similar environments. Classical mapping and planning baseline resulted in 45.20% planning performance as compared to 66.16% of SPT in the Habitat navigation test set. This is consistent with the results in [8] who also find that classical mapping and planning is not very effective given only RGB input.

---

> > ### Author Response · Authors · 2020-11-25
> > **Response to AnonReviewer1 (cont.)**
> >
> > References:
> >
> > [1] Bansal et al. Combining Optimal Control and Learning for Visual Navigation in Novel Environments. CoRL, 2020
> >
> > [2] Tamar et al. Value Iteration Networks. NeurIPS, 2016
> >
> > [3] Lee et al. Gated Path Planning Networks. ICML, 2018
> >
> > [4] Kitaev et al. Reformer: The Efficient Transformer. ICLR, 2020 (https://arxiv.org/pdf/2001.04451.pdf)
> >
> > [5] Dai et al. Transformer-XL: Attentive Language Models Beyond a Fixed-Length Context. https://arxiv.org/abs/1901.02860
> >
> > [6] Shen et al. Efficient Attention: Attention with Linear Complexities. https://arxiv.org/abs/1812.01243
> >
> > [7] Anonymous. LambdaNetworks: Modeling long-range Interactions without Attention. https://openreview.net/forum?id=xTJEN-ggl1b
> >
> > [8] Mishkin et al. Benchmarking Classic and Learned Navigation in Complex 3D Environments. https://arxiv.org/pdf/1901.10915.pdf
> >
> > [9] Chen et al. Learning to Plan in High Dimensions via Neural Exploration-Exploitation Trees. ICLR, 2020
> >
> > [10] Karkus et al. Qmdp-net: Deep learning for planning under partial observability. NeurIPS, 2017
> >
> > [11] Hu et al. Revisiting single image depth estimation: Toward higher resolution maps with accurate object boundaries. WACV, 2019

---

### Decision · Program_Chairs · 2021-01-07
**Final Decision**

**Decision:**

Reject

**Comment:**

This paper proposes to jointly learn a mapper and planner for navigation or manipulation in a 2D space represented by an MxM grid, with the mapper taking raw observations as inputs and producing a 2D MxM occupancy and goal location map, and the planner -- pretrained on generic 2D maps of same size MxM -- produces an MxM action distance image (the plan). The whole system is trained end-to-end, with the mapper trained on the specific navigation or manipulation task, and the planner frozen. The Spatial Planning Transformer network can predict the MxM action plans faster than baselines (Value Iteration Networks and Gated Path Planning Networks) because of the attention mechanism in the transformer architecture, as opposed to the local information propagation through the convolutional encoding of the Bellman equation in VIN or GPPN. The differentiable approach is motivated by exploiting regularities in 2D maps and a faster inference time (as opposed to classical Dijkstra planners or VIN / GPPN), and is demonstrated on a simple room navigation task in the Gibson environment.

Reviewers have praised how a simple idea (transformer-based planning à la VIN) can be applied as a unifying approach to 2 DOF manipulation and 2D navigation, and the out-of-sample evaluation. The critique was about:
* the lacunary explanation of the mapper (corrected by the authors),
* the limits of non-recurrent mappers that cannot handle occlusions in the observations when building maps,
* confusion about where the full MxM image of action labels (necessary for supervision) can come from (the authors added experiments with sparse labels),
* claims about this approach being preferable to A* (countered by the authors, who conducted extensive experiments at the request of R4),
* focus on two problems at once (mapping and planning) rather than more ablation analysis of the planner, with a potential comparison to Active Neural SLAM,
* missing evaluation using navigation-specific metrics, such as SPL.

The authors have a point in their defense of differentiable / learning-based methods for planning (including VIN and GPPN) as opposed to classical planning such as A*, and there is value in investigating how an end-to-end differentiable method for mapping and planning could be designed. At the same time, several reviewers raised concerns about scalability and about the pertinence of combining two learnable modules (mapper and planner) rather than investigating and demonstrating the advantage of transformers for planning.

Given these reviews, rebuttal, and remaining concerns, I am sorry to reject this paper. I hope that with the proposed modifications it will be quickly accepted at another venue.